# Solvothermal In-Situ Synthesis of MIL-53(Fe)@Carbon Felt Photocatalytic Membrane for Rhodamine B Degradation

**DOI:** 10.3390/ijerph20054571

**Published:** 2023-03-04

**Authors:** Shuyan Yu, Huiying Zhang, Congju Li

**Affiliations:** 1School of Energy and Environmental Engineering, University of Science and Technology Beijing, Beijing 100083, China; 2Beijing Key Laboratory of Resource-Oriented Treatment of Industrial Pollutants, Beijing 100083, China; 3Energy Conservation and Environmental Protection Engineering Research Center in Universities of Beijing, Beijing 100083, China

**Keywords:** MIL-53(Fe)@carbon felt, photocatalytic, degradation, rhodamine B

## Abstract

In this study, MIL-53(Fe) was innovatively incorporated into carbon felt (CF) by growing in-situ using the solvothermal method. MIL-53(Fe)@carbon felt (MIL-53(Fe)@CF) was prepared and used for the degradation of rhodamine B (RhB). As a new photocatalytic membrane, MIL-53(Fe)@CF photocatalytic membrane has the characteristics of high degradation efficiency and recyclability. Influence of various parameters including MIL-53(Fe)@CF loading, light, electron trapper type, and starting pH on RhB degradation were investigated. The morphology, structure, and degradation properties of MIL-53(Fe)@CF photocatalytic membrane were characterized. Corresponding reaction mechanisms were explored. The results indicated that pH at 4.5 and 1 mmol/L H_2_O_2_, 150 mg MIL-53(Fe)@CF could photocatalytically degrade 1 mg/L RhB by 98.8% within 120 min, and the reaction rate constant (k) could reach 0.03635 min^−1^. The clearance rate of RhB decreased by only 2.8% after three operations. MIL-53(Fe)@CF photocatalytic membrane was found to be stable.

## 1. Introduction

Water pollution has long been a source of concern for the environment, with dyeing wastewater accounting for a considerable proportion of industrial wastewater discharge. Dyeing wastewater contains residual dyes in addition to additives, acids, alkalis, and other chemicals. Aromatic chemical molecules and heavy metal elements such as chromium and cadmium, which are toxic and dangerous to the human body, can enter organisms through the biological chain [1].

The pollution of some toxic organic dyes into the water environment has become increasingly serious with the development of organic dyes in all walks of life. Some scholars have prepared porous Cu/C nanocomposites for photocatalytic degradation of methyl violet (MV), and its photocatalytic activity is significantly higher than that of other Cu/C nanocomposites. The degradation of MV can reach 96% after being cycled four times [2]. Among these traditional organic dyes, RhB and Con T (ST) represent a major threat to human health since they include carcinogenic functional groups such as benzene ring, naphthalene, and benzoquinone [3]. As a result, if these issues are not addressed in an efficient and accurate manner, they will pose a major threat to the human living environment [4]. Scholars have studied the removal of RhB from the environment. Wang et al. [5] created a new coordination polymer, Zn(bpy)L (BUC-21). BUC-21 has high photocatalytic activity in the degradation of organic dyes, e.g., 10 mg/L RhB could be degraded more than 80% in 30 min. A Co-MOF with a 2D morphology (BUC-92) was prepared. BUC-92 has faster electron transfer efficiency, which is more helpful for the degradation of RhB [6].

Adsorption, oxidation, and photocatalysis are the most commonly studied methods for eliminating dye molecules [7]. Photocatalysis is an interdisciplinary integration of energy, materials, life, environment, information, and other disciplines. It can directly convert solar energy into chemical energy, making it an effective strategy for clean renewable energy and environmental purification. Some scholars [8] discovered that when TiO_2_ is used as a photocatalyst and exposed to UV irradiation, organic matter can be oxidized and degraded into chemicals such as H_2_O and CO_2_. Since then, photocatalytic degradation technology has attracted much attention. Photocatalysis is widely regarded as one of the most attractive wastewater treatment processes. However, photocatalysis faces two weaknesses that limit its development in practical industrial applications: firstly, photocatalysts mainly absorb ultraviolet and utilize rarely visible light; secondly, the photogenerated carrier recombination rate is high and the quantum efficiency is low [9].

Metal-organic framework materials (MOFs) are new photocatalysts that are rapidly becoming a research hotspot. MOFs have excellent adsorption performance due to their large specific surface area and many active adsorption sites, which allows MOFs to improve the adsorption efficiency of target adsorbents by adjustment [10]. MOFs have excellent adsorption properties against organic poisons in wastewater, such as antibiotics, anti-inflammatory, antipsychotics, organophosphorus pesticides and other drugs [11]. Qin et al. [12] synthesized a new microporous 2D Co-based MOF. It is used for CO_2_ adsorption, and the adsorption capacity can reach 2.88 mmol/g. MOFs also act as a carrier for loading drugs because of its large specific surface area [13]. As MOFs are typically in powder form, they can be easily lost, scattered and hard to recover during practical applications, thus necessitating the use of other substances to make it more stable. Polymer composites can enhance the properties of MOFs by stabilizing the skeleton [14] or enhancing the absorption of the desired analyte.

MIL-53(Fe) is a three-dimensional porous solid MOF formed by the cross-linking of -Fe-O-Fe-O-Fe-O-Fe- and 1,4-benzoic acid in infinite one-dimensional connections [15]. Under light irradiation, electron excitation occurs in MIL-53, followed by electron transfer, showing high photocatalytic activity. However, MIL-53(Fe) catalyst is in a powder state and difficult to recover from water, which limits its development in the field of photocatalysis. Carbon felt is a carbon fiber material with good conductivity in photocatalysis. It can rapidly transfer electrons, effectively improve the formation efficiency of electron holes, and improve photocatalytic performance [16]. Therefore, carbon felt was used as the base material of MIL-53(Fe), so that MIL-53(Fe) grew in-situ on the nitric acid-modified carbon felt, finally facilitating the reuse and recycling of MIL-53(Fe).

At present, there are many methods to combine MOFs with fabrics, such as electrospinning [17], solvents, thermal atomic layer deposition [18], active surface fabric method [19], and hot pressing method [20]. This study aimed to grow and synthesize MIL-53(Fe) on modified carbon felt in-situ by solvothermal reaction. MIF-53(Fe) can increase the specific surface area and MIF-53(Fe) is more conducive to excited electron transfer, CF has excellent electrical conductivity and can promote electron transfer, so that the photocatalytic efficiency was enhanced. MIL-53(Fe)@CF overcomes the shortcomings of traditional granular catalysts, which are difficult to recovery and easy to agglomerate. Photocatalytic degradation of rhodamine B (RhB) by MIL-53(Fe)@CF was studied, and the effects of light exposure, hole trap species, initial pH value of the system, and catalyst dosage on photocatalytic degradation of RhB were investigated. Corresponding reaction mechanisms were explored. Stability of MIL-53(Fe)@CF photocatalytic membrane was verified, which provided a theoretical and experimental basis for the practical application of MIL-53(Fe)@CF as a composite photocatalytic membrane in dyeing wastewater.

## 2. Materials and Methods

### 2.1. Materials

Ferric chloride hexahydrate (FeCl_3_·6H_2_O, 99.9%), hydrogen peroxide (H_2_O_2_, 99.9%), potassium bromate (KBrO_3_, 99.9%), ammonium persulfate ((NH_4_)_2_S_2_O_8_, 99.9%), terephthalic acid (H_2_BDC, 99.9%) and N, N-dimethylformamide (DMF, 99.9%) were purchased from Aladdin. Carbon felt was purchased from Jingzhou Haote New Materials Co., Ltd. (Jingzhou, China).

### 2.2. Preparation of the Modified Carbon Felt and MIL-53 (Fe)@CF

Due to the weak adhesion of MIL-53(Fe) to the surface, the load capacity of MIL-53(Fe) of carbon felt is low. To solve this problem, 30% nitric acid was used for pretreatment to improve the adhesion and increase the loading capacity of MIL-53(Fe). The treated MIL-53(Fe)@ carbon felt was soaked in methanol to ensure sufficient removal of residual loose substances for subsequent experiments. The commercial carbon felt was cut into sizes of 2.0 cm × 2.0 cm × 0.3 cm, then washed and soaked in acetone for 24 h. After rinsing with deionized water (DI water), it was soaked in nitric acid (concentration was 30%) for 24 h. After rinsing with DI water, it was dried in an oven at 80 °C for 12 h to obtain carbon felt modified with nitric acid.

MIL-53(Fe)@CF was synthesized as follow; first 0.676 g (2.5 mmol) FeCl_3_·6H_2_O and 0.415 g (2.5 mmol) terephthalic acid (H_2_BDC) were dissolved in 10 mL N, N-Dimethylformamide (DMF). The solution was poured into a solvent-hot pressure vessel lined with Teflon. Carbon felt was then added and dried continuously for 12 h in an oven at 130 °C. After drying, it was washed with DI water and immersed in methanol for 24 h. The treated carbon felt was dried in an oven at 80 °C for 12 h to obtain in-situ growth of MIL-53 (Fe)@CF photocatalytic membrane.

### 2.3. Characterization of the MIL-53 (Fe)@CF

The morphology and composition of the membrane were characterized by an scanning electron microscope (SEM; NOVANNANOSEM450) with an accelerated electrical charge of 10 keV. X-ray diffraction (XRD; Bruker D8) was performed with Cu-Kα radiation in the scanning angle range of 5–30° at 150 mA, 40 kV. Fourier transform infrared (FT-IR) used the nicolet-6700 for analysis in the wavenumber range of 400 to 4000 cm^−1^.

### 2.4. Experimental Method

In this study, the photocatalytic oxidation property of MIL-53(Fe)@CF was investigated by the degradation of RhB aqueous solution under visible light radiation. The experiment was carried out in a cylindrical Pyrex vessel reactor using a 300 W Xe arc lamp as the sample light source, with the lamp current set at 15A when illuminated. A certain amount of MIL-53(Fe)@CF was added to 100 mL 1 mg/L RhB solution (pH = 4.5), the mixed solution was stirred for 80 min without magnetic light, the catalyst reached adsorption equilibrium. The sample solution was sampled every 20 min within 2 h of light degradation, and the fall-off MIL-53(Fe) was separated with the sample filtered by centrifugation. After centrifugation, the absorbance of RhB clear liquid was measured by ultraviolet visible spectrophotometer. the degradation of RhB could be judged by the absorbance of the sample in 450 nm to 650 nm. Then the degradation rate-time curve of RhB under the experimental conditions was obtained.

## 3. Result and Discussion

### 3.1. Characterization of the In-Situ Growth of MIL-53(Fe)@CF Photocatalytic Membrane

The morphology of the sample was observed by SEM (Figure 1). Carbon felt without synthetic MIL-53(Fe) (Figure 1a) has a smooth surface and no nanoparticles growth. However, MIL-53(Fe) was grown in-situ on the surface of carbon felt (CF) using the solvothermal method (Figure 1b), in which 30% nitric acid was used for pretreatment to improve the adhesion and increase the loading capacity of MIL-53(Fe). This synthetic method leads to the formation of MOF-based nano-composites that uniformly cover the surface of carbon felt fibers (dimensions of 300–500 nm). MIL-53(Fe) is deposited on the carbon felt surface because the carboxylate ligand provides an O_6_-coordination environment around the iron center. Previous research has revealed a catalytic iron oxide site for MIL-100(Fe) [21,22].

XRD analysis is performed to reveal the composition and crystal phases of the in-situ growth of MIL-53(Fe)@CF. As shown in Figure 2a, the characteristic peaks at 2θ = 9.28°, 12.9°, 17.5°, and 26.02° are attributed to the (200), (110), (111), and (220) crystal planes of MIL-53(Fe). The amorphous peak of carbon felt is at 2θ = 25.5°, which could be well indexed to MIL-53(Fe)@CF. MIL-53(Fe)@CF photocatalytic membrane was successfully synthesized under these experimental parameters.

FT-IR analysis was used to determine the structure of the MIL-53(Fe)@CF (Figure 2b). The FT-IR was measured at a wavelength of 400 cm^−1^ to 4000 cm^−1^, but only the wavelength of 500 cm^−1^ to 2000 cm^−1^ shows the significant peaks, so 500 cm^−1^ to 2000 cm^−1^ was selected to show in Figure 2b. The functional groups and chemical bond types of substances can be represented by infrared spectroscopy. The carbon felt is composed of elemental carbon that is without specific peak. MIL-53(Fe)@CF photocatalytic membrane contains several distinct peaks in the 1700 cm^−1^ to 1400 cm^−1^ range. The telescopic vibration of the carboxyl group is responsible for the peak [23]. The absorption peak at 1594 cm^−1^ is produced by the ligand carboxyl with the metal core. The absorption peaks at 1298 cm^−1^ and 750 cm^−1^ are attributed to the symmetrical vibration of carboxyl group [24] and the carbon-hydrogen bonds of the benzene ring [25]. In addition, the telescopic vibration band at 552 cm^−1^ is related to the presence of Fe-O bonds, which is due to the formation of metal oxygen bonds between carboxyl groups in 1, 4-benzenedicarboxylic acid, and iron (III) [26].

### 3.2. Optimal MIL-53(Fe)@CF Dosage and Dark Reaction Time for RhB Removal

In this study, the effects of 10, 50, 100, 150, and 200 mg of catalyst on 100 mL of 1 mg/L RhB degradation performance were explored. After being stirred for 80 min for adsorption equilibrium, the solution was exposed to light irradiation for two hours. The degradation rate of RhB was found to remain stable at 150 mg of catalyst, thus the optimum dosage was 150 mg. An aqueous solution with gradient concentration RhB is prepared in advance to prepare UV absorption spectra of the standard solution (Appendix A). The absorbance and concentration curves are shown in Appendix A.

To produce an adsorption and desorption equilibrium, the mixed solution was magnetically stirred for 80 min while no light was provided. After visible light irradiation, we removed 1 mL of sample at predetermined intervals and filtered the exfoliated MIL-53(Fe). At a maximum RhB absorption wavelength of 554 nm, the residual RhB concentration in the supernatant after upper centrifugation was determined using a UV/VIS spectrophotometer.

The degradation efficiency of the catalyst sample is expressed in η, which is calculated as:η = (1 − C/C_0_) × 100%(1)
where η represents the degradation efficiency (%) of the catalyst as the concentration after the pollutant reaction, C_0_ the pollution Initial concentration.

The degradation degree of the catalyst sample could be calculated as:Degradation degree = ln(c_t_/c_0_)(2)
where c_t_ represents the absorbance after the pollutant reaction, c_0_ the pollution initial absorbance, the degradation rate constant of RhB can be calculated by the ln(c_t_/c_0_) − t.

Concentration of RhB in the reaction under light irradiation. After 80 min of adsorption, the RhB concentration in the solution decreased from 1 mg/L to 0.86 mg/L and remained steady (Figure 3). As a consequence, in-situ growth of MIL-53(Fe)@CF photocatalytic membrane takes 80 min to reach adsorption equilibrium in RhB aqueous solution. In following experiments, stirring with a magnetic stirrer for 80 min under relatively lightless conditions is required before the photocatalytic reaction occurs.

### 3.3. Effects of Catalysts, Light Irradiation, and Electron Trapping Agents on Photocatalytic Degradation of RhB

In the experiment, the presence of a single condition among the three conditions of light irradiation, electron trapping agent, and catalyst was controlled. Experimental results of controlling a single condition were compared with the experimental results when all three conditions existed at the same time.

If there is a lack of light or catalyst, the concentration of RhB does not change significantly after 120 min of reaction (Figure 4). Removal rate increases slightly in the absence of the electron trap, but it is still deficient. When there is light, catalyst, and H_2_O_2_, the degradation rate of RhB reaches 98.8% in 120 min. This means that photogenerated electrons and electron holes can only be produced on the surface of MIL-53(Fe)@CF photocatalytic membrane under light irradiation. However, electron holes quickly recombine, resulting in low degradation efficiency. When adding H_2_O_2_, it binds with electrons, allowing more electron holes to participate in the reaction, generating photocurrent and increasing the reaction rate. Therefore, RhB is degraded by MIL-53(Fe)@CF photocatalytic membrane in the presence of light irradiation, H_2_O_2_, and catalyst.

### 3.4. Effect of Electron Trap Species on Photocatalytic Oxidation Dye Reaction

Suitable organic matter is usually added to the reaction of routine photocatalytic oxidation dyes as an electron trapping agent. The organic matter can increase the photogenerated charge separation efficiency in the photocatalytic process. In this experiment, three organic compounds were selected as electron trapping agents: H_2_O_2_, KBrO_3_, and (NH_4_)_2_S_2_O_8_. The pH was adjusted (pH at maximum adsorption of MOF dyes) using dilute NaOH and HCl solution.

KBrO_3_ is used as an electron trapper. RhB in the solution is degraded by 44.5% after 120 min of light. Under the same conditions as adding (NH_4_)_2_S_2_O_8_ as an electron trap, RhB is degraded by 91.4%. H_2_O_2_ shows stronger degradation performance as an electron trapping agent, with a degradation efficiency of 98.8% (Figure 5). Therefore, H_2_O_2_ can significantly accelerate the reaction rate of photocatalytic degradation of RhB. In the follow-up study of this paper, H_2_O_2_ was selected as an electron capture agent to participate in the experiment.

Table 1 lists the reaction rate constants of H_2_O_2_, KBrO_3_, and (NH_4_)_2_S_2_O_8_ as electron trapping agents. The reaction kinetic curves when H_2_O_2_, KBrO_3_ and (NH_4_)_2_S_2_O_8_ were added follow the first-order kinetic law. Appendix A shows the absorbance of RhB degradation process. The absorbance at 554 nm was selected to make the ln(c_t_/c_0_)—t figure. Appendix A shows the kinetic curve of the reaction when H_2_O_2_ is added, and the photocatalytic reaction rate is significantly increased, H_2_O_2_ increases the reaction rate constant. The reaction rate constant of the photocatalytic reaction is 0.03635 min^−1^ when H_2_O_2_ is used as an electron trapping agent. Appendix A shows the kinetic curve of the reaction when ((NH_4_)_2_S_2_O_8_) is added, (NH_4_)_2_S_2_O_8_ has a specific effect on the photocatalytic reaction, and the reaction rate constant of the photocatalytic reaction is 0.020115 min^−1^. It is the kinetic curve of the reaction when KBrO_3_ is added which has a limit effect on the photocatalytic reaction, and the reaction rate constant of the photocatalytic reaction is 0.00457 min^−1^ (Appendix A).

### 3.5. Effect of Initial pH of the Reaction Solution on the Photocatalytic Oxidation of Dyes

The photocatalytic reaction was significantly influenced by the pH. The pH of the experiment was limited between 3.5 to 5.5. In order to enhance the efficiency of photocatalytic degradation, H_2_O_2_ was used in the experiment as an electron trapping agent to slow down electron-hole recombination. Dilute NaOH and HCl solutions were used to adjust pH. The pH of the solution was stable at 3.5, 4.5, and 5.5.

After 120 min of light irradiation, the pH of the solution system was 3.5, 4.5 and 5.5, the degradation rate of RhB in the solution is 84.8%, 98.8% and 68.5% (Figure 6). The degradation efficiency of pH at 4.5 is significantly higher than pH at 3.5 and 5.5. According to Equations (4) and (5), this may be because when the pH was above 4.5, the formation of ·OH and ·O_2_^−^ was inhibited. When the pH was below 4.5, according to Equation (5), OH^−^ may be consumed, which reduced the concentration of ·O_2_^−^ as a reactant. According to Equation (8), the low pH may inhibit the formation of ·OH, which reduced the concentration of free radicals in the solution. When pH was at 3.5 and 5.5, the anion and cation concentration in the solution changed, and protonation or deprotonation may have occurred on the surface of the catalyst, which may have bound back to some of the active sites of the reaction and which affected the adsorption of RhB by the catalyst.

Appendix A show that when the pH was at 3.5, 4.5, and 5.5, the kinetic curve of all reactions conforms to the law of first-order kinetics (k). The k was calculated according to the absorbance of Appendix A. Table 2 summarizes the reaction rate constants at different pH. The reaction rate constant k at 4.5 can reach 0.03635 min^−1^, which is 3.6 times the reaction rate constant k at 5.5. The reaction rate constant reflects the rate at which RhB is degraded, and it can be seen from the UV-Vis curve that RhB is most degraded when the pH at 4.5. Therefore, the pH at 4.5 was selected as the pH of the reaction in subsequent experiments.

### 3.6. MIL-53(Fe)@CF Stability

Photocatalytic membrane stability experiments were performed under optimized reaction conditions. The MIL-53(Fe)@CF membrane was separated by filtration after first reaction, and the carbon fiber membrane was rinsed repeatedly with a large amount of deionized water until the solution became clear. After filtration, the RhB adsorbed on the carbon fiber membrane was separated using ultrasonic vibration of deionized water for 20 min through three washes. Following that, MIL-53(Fe)@CF was taken out and dried. Dried carbon fiber film was used as the photocatalyst required for the reaction, and the photocatalytic experiment was repeated three times. Measuring the degradation removal rate of RhB in each solution occurred separately using UV spectrophotometer.

The degradation efficiency of RhB decreased from 98.8% to 96.0% after MIL-53(Fe)@CF was repeated three times (Figure 7). This may indicate that the catalyst characteristics are relatively stable, and high photocatalytic degradation efficiency can still be achieved after three photocatalytic experiments.

### 3.7. Mechanism of MIL-53(Fe)@CF Photocatalytic Degradation of RhB

The photocatalytic process of RhB degradation may involve an electron transition from the valence band (VB) to the conduction band (CB) and produce photogenerated electron-hole pairs [27]. Some photogenerated holes (h^+^) or electrons (e^−^) can migrate to the surface of the catalyst, then recombine and release in a high-energy form. The majority of the e^−^-h^+^ will quickly recombine in the semiconductor; therefore, oxidation is lost during migration to the surface. Electron trap H_2_O_2_ can combine with e^−^, allowing more h^+^ to participate in the reaction.

O_2_ and e^−^ react can produce ·O_2_^−^ which can convert into ·OH. H^+^ moves to the surface of the catalyst and can react with H_2_O to form ·OH. These processes can produce a lot of ·O_2_^−^ and ·OH. ·O_2_^−^ and ·OH have strong oxidation, and can degrade RhB into CO_2_ and H_2_O [28]. At the same time, h^+^ can also directly degrade RhB to CO_2_ and H_2_O. As a result, the organic dyes wastewater is decolorized. As shown in Figure 8, when photogenerated electron holes are generated on MIL-53(Fe), the orbital energy of MIL-53(Fe) is 2.72 eV. It is then combined with CF, the photogenerated carrier transfer efficiency of MIL-53(Fe) is enhanced, and then the photocatalytic performance of MIL-53(Fe)@CF is improved. By adding hydrogen peroxide (H_2_O_2_), potassium bromate (KBrO_3_), and ammonium persulfate ((NH_4_)_2_S_2_O_8_) electron acceptors, the recombination of electron-hole pairs can be inhibited, thereby improving the photocatalytic degradation of RhB by generating more free radicals. Therefore, the introduction of an external electron catcher can enhance the performance of MIL-53(Fe) photocatalysts.

The above photocatalytic degradation of RhB can be summarized as following:MIL-53(Fe) + hv→e^−^(MIL-53(Fe)) + h^+^(MIL-53(Fe))(3)
e^−^(MIL-53(Fe)) + O_2_→·O_2_^−^(4)
O_2_^−^ + H_2_O→HO_2_ + OH^−^(5)
HO_2_ + H_2_O→·OH + H_2_O_2_(6)
H_2_O_2_→2·OH(7)
h^+^(MIL-53(Fe)) + H_2_O→·OH + H^+^(8)
·OH + RhB→CO_2_ + H_2_O(9)
h^+^(MIL-53(Fe)) + RhB→CO_2_ + H_2_O(10)

As reported by other studies, the main degradation intermediates of RhB are determined by GC/MS and LC/MS, and on the basis of previous studies [31,32,33,34,35], the results of its degradation intermediates are summarized as shown in Figure 9. The photocatalytic degradation steps of RhB are N-deethylation, chromophore clearage, opening-ring, and finally mineralized as CO_2_ and H_2_O.

## 4. Conclusions

In this work, MIL-53(Fe) was grown in-situ on nitric acid-modified carbon felt and used in degradation of RhB by photocatalysis. Carbon felt has good conductivity in photocatalysis, as it can make rapid transfer of electrons and effectively improve the formation efficiency of electron holes, thus improve photocatalytic performance. In this study, MIL-53(Fe)@CF photocatalytic membrane was synthesized, RhB was catalytically oxidized under light irradiation, and the effects of the presence or absence of light, electron trap type, reaction system pH, and catalyst dosage were systematically explored. Experiments had shown that when light and H_2_O_2_ were added, the photocatalytic ability of was significantly increased. At pH 4.5 and H_2_O_2_ concentration of 1 mmol L^−1^, 150 mg MIL-53(Fe)@CF can photocatalytically degrade 1 mg/L of RhB by up to 98.8% in 120 min. This study effectively fixed MOFs by growing MIL-53 in-situ on carbon felt, which provided a reference for MOFs powder fixation.

## Figures and Tables

**Figure 1 ijerph-20-04571-f001:**
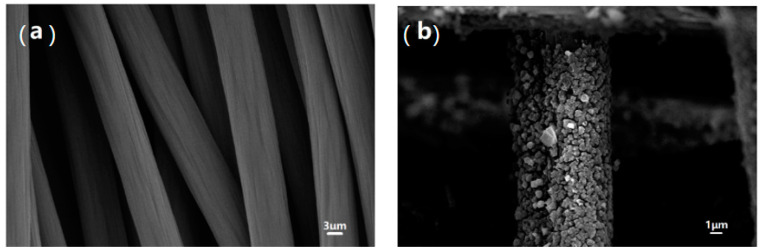
SEM images of unmodified carbon felt (**a**) and in-situ grown MIL-53(Fe)@CF photocatalytic membrane (**b**).

**Figure 2 ijerph-20-04571-f002:**
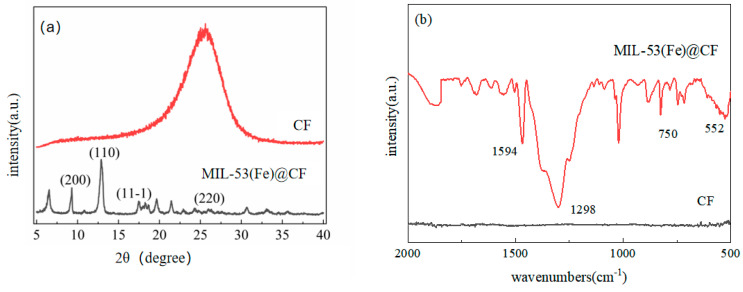
MIL-53(Fe) @CF in situ growth of XRD (**a**) and FT-IR (**b**).

**Figure 3 ijerph-20-04571-f003:**
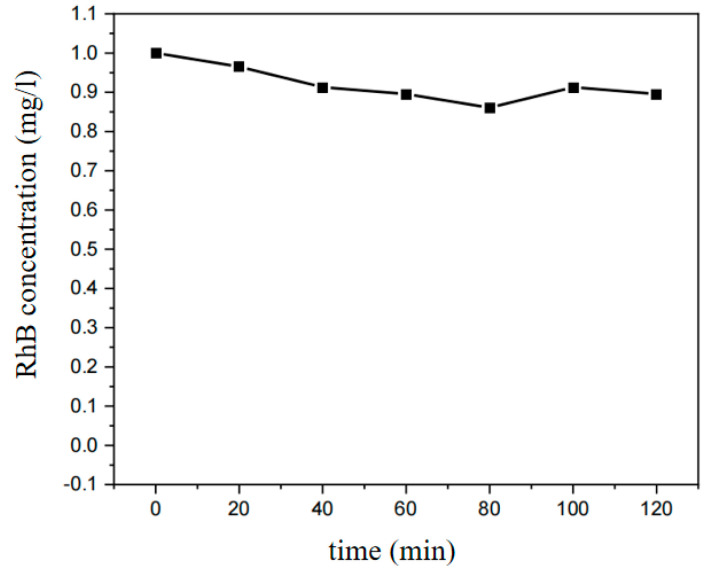
Adsorption experiment of MIL-53(Fe)@CF photocatalytic membrane to RhB under no light irradiation (pH at 4.5, MIL-53(Fe)@CF photocatalytic membrane at 150 mg, RhB 1 mg/L).

**Figure 4 ijerph-20-04571-f004:**
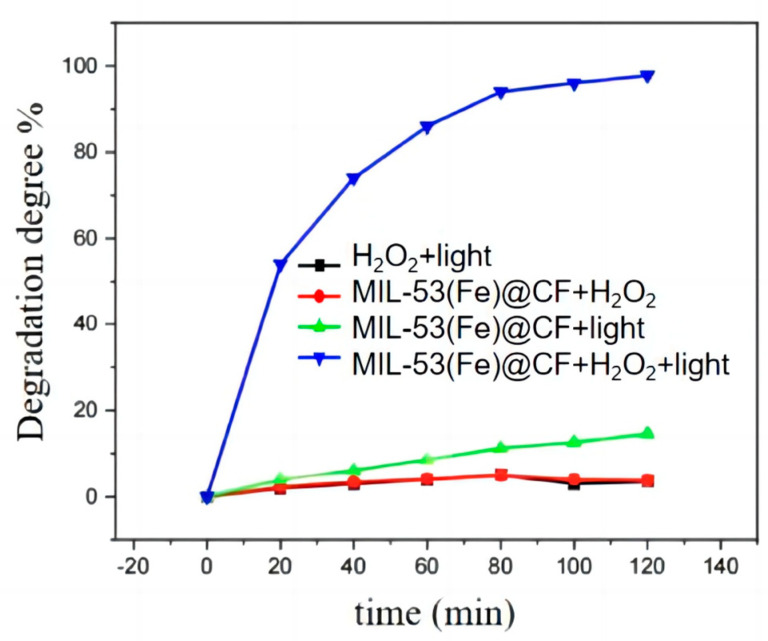
Effects of different conditions on photocatalytic oxidation of RhB. (Reactor: pH at 4.5, MIL-53(Fe)@CF photocatalytic membrane 150 mg, RhB 1 mg/L, H_2_O_2_ 1 mmol/L).

**Figure 5 ijerph-20-04571-f005:**
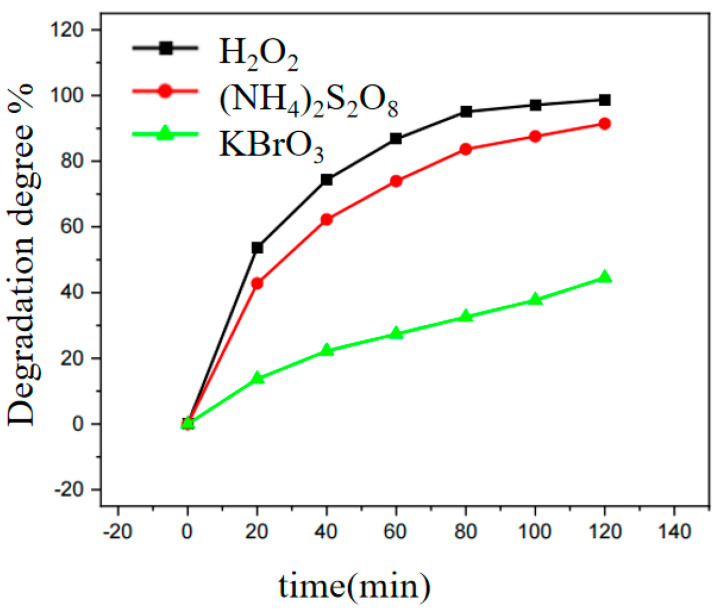
Effect of different electron trapping agents on photocatalytic oxidation of RhB. (Reactor: pH at 4.5, MIL-53(Fe)@CF photocatalytic membrane 150 mg, RhB 1 mg/L, H_2_O_2_ 1mmol/L).

**Figure 6 ijerph-20-04571-f006:**
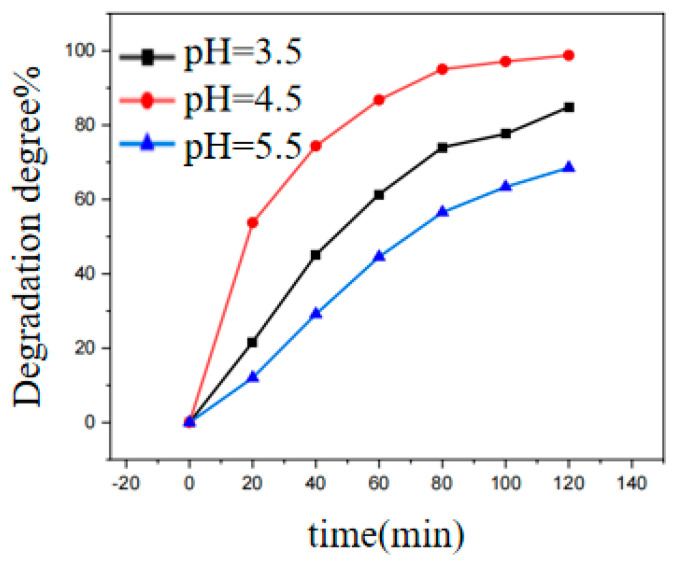
Effects of different pHs on photocatalytic oxidation of RhB. (Reactor: MIL-53(Fe)@CF photocatalytic membrane 150 mg, RhB 1 mg/L, H_2_O_2_ 1 mmol/L).

**Figure 7 ijerph-20-04571-f007:**
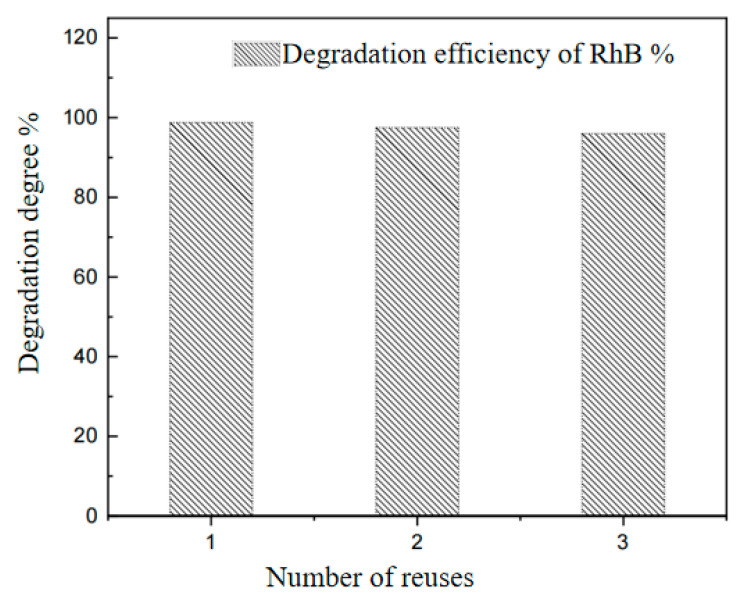
The removal rate of RhB in recycling. (Reactor: pH at 4.5, MIL-53(Fe)@CF photocatalytic membrane 150 mg, RhB 1 mg/L, H_2_O_2_ 1 mmol/L).

**Figure 8 ijerph-20-04571-f008:**
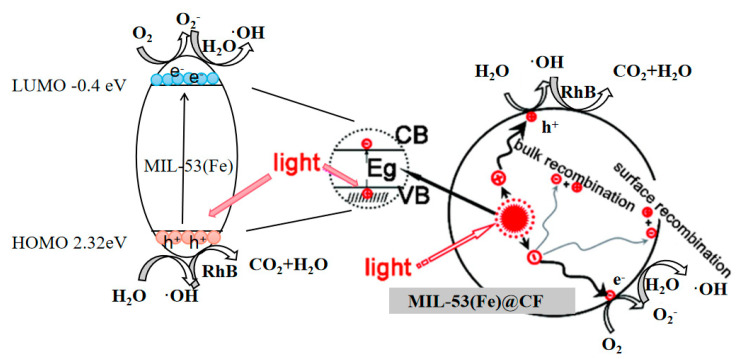
MIL-53(Fe)@CF membrane photocatalytic degradation of RhB [29,30].

**Figure 9 ijerph-20-04571-f009:**
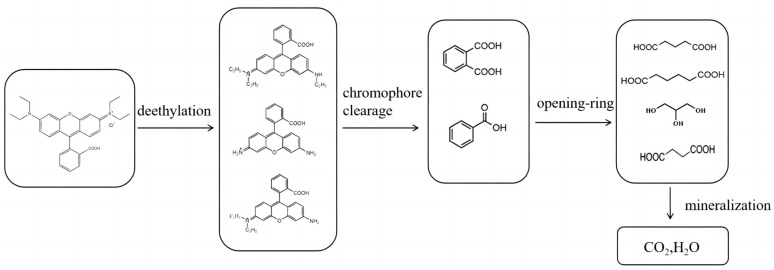
The degradation pathway of RhB [31,32,33,34,35].

**Table 1 ijerph-20-04571-t001:** Reaction rate constants for different types of electron capture agents.

Types of Electronic Capture Agents	Reaction Rate Constant K (min^−1^)	R^2^
(NH_4_)_2_S_2_O_8_	0.02011	0.9914
KBrO_3_	0.00427	0.8883
H_2_O_2_	0.03635	0.996

**Table 2 ijerph-20-04571-t002:** Reaction rate constants for different pH.

pH	Reaction Rate Constant k (min^−1^)	R^2^
3.5	0.01591	0.992
4.5	0.03635	0.996
5.5	0.01019	0.992

## Data Availability

Data is contained within the article or Appendix A. The data presented in this study are available in insert article or Appendix Ahere.

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
