# Peer review of "Solvothermal In-Situ Synthesis of MIL-53(Fe)@Carbon Felt Photocatalytic Membrane for Rhodamine B Degradation"

_ijerph, 2023, doi:10.3390/ijerph20054571_

Round 1

Reviewer 1 Report

Reviewers' Comments:

This study reports that the degradation efficiency of Rhodamine B (RhB)  photocatalytic degradation by MIL-53(Fe)@carbon felt (MIL-53(Fe)@CF). The findings provided more insight that RhB is efficiently degraded by MIL-53(Fe)@CF photocatalytic system. The results indicated that pH at 4.5 and 1mmol/L H2O2, 150 mg MIL-53(Fe)@CF could photocatalytically degrade 1 mg/L RhB by 98.8% within 120 min. There are still some problems in the work. Therefore, in order to further improve the content of the work, the following points should be taken into account. Overall, this manuscript meet the standard of International Journal of Environment Research and Public Health, I recommend it to be published after minor revision.

1. How does the degradation of RhB compare with other ways?

2. Fig. 3-7 (a) of the supplementary document not be mentioned in the manuscript and should be explained.

3. Whether the purchased carbon felt materials are further processed should be clearly stated.

4. Does the loading amount of MIL-53(Fe) have an effect on the catalytic performance?

5. Some MOFs for photocatalytic RhB degradation were reported, like Chinese Journal of Catalysis 38 (12), 2141-2149, CrystEngComm 24, 5557–5561. The authors can cite them in the revised manuscript.

Reviewer 2 Report

The innovation and justification of this work should be better explored in the introduction.

 The methodology of photocatalysis must be included in the manuscript.

The authors mention the photodegradation mechanism, but only the degradation kinetics with some parameters was presented. The mechanism involves, for example, the part of photoproducts, which shows that there was the degradation of the model molecule and not just a follow-up to a decrease in the concentration of the pollutant. 

Section 3.2 can be placed in the supplemental material.

Figure 4 should show experiments with only hydrogen peroxide and the dye. In the Figure, it should be written that there is a photometer. All subtitles must be redone. 

My main concern is that the absorbance shown in Figure 3S is very low, almost at the limit of the equipment, in the noise range, which can lead to an error in the determination of kinetics. Do this experiment at a concentration where the absorbance is close to at least 0.8. It is not presented in an explanatory way of how the recycling was done or the conditions that were made. It must be placed in the methodology and all other parameters. 

There needs to be standardization of Figures in writing. The text must be revised entirely. 

The mechanism (Figure 8) presented is fundamental, and there is no specification of its material.

Reviewer 3 Report

Although the title of manuscript „Solvothermal in-situ synthesis of MIL-53(Fe)@carbon felt photocatalytic membrane for rhodamine B degradation“ suggests that it should be focused on the synthesis and characterization of a new type of photocatalytic material, my expectations were not fulfilled. In addition to this content weakness, I consider the language level of the text to be a major obstruction to the publication of the submitted manuscript, not only the very low grammatical level of English, but above all the choice of words (terms) and sentence structure. Some sentences do not contain a verb and thus make no sense. If the publication of the manuscript should be considered, its overall language revision is necessary first.

Next, I present the main factual comments on the submitted manuscript:

Labels of images and graphs, names of physical quantities used both in graphs and in text, etc., require special attention. For example:

Figure 1 - SEM images instead of SEM plot

Figure 2 –the x-axis should be named wavenumbers and not wavelength

Figure 4 and others – the y-axis does not represent the rate of degradation but the degree of degradation, I suppose

Chapter 3.1 Characteriaztion of the in-situ growth of MIL-53(Fe)@CF photocatalytic membrane should be completely revised. The presented results (SEM, XRD and FTIR) do not prove in any way that MIL-53(Fe) is fixed on carbon fibers. The presence of carbon could be proven, for example, by Raman spectroscopy. Citations 15 and 16 refer to articles on the structure of other types of MOFs, and the results are not so easily comparable. The interpretation of FTIR spectra is very general. I recommend supplementing the structural formulas of MIL-53(Fe) and MIL-53(Fe)@CF and describing the measured FTIR spectra on them.

Moreover, any characterization of the electrochemical properties is completely missing.

In the chapters comparing the efficiency of the photocatalytic degradation of RhB (3.2. - 3.6.), it is first necessary to clearly explain which parameters were changed, in which order, and also which parameters, on the other hand, remain unchanged and at which values. From this point of view, it is also necessary to think about the order of the individual chapters.

Reviewer 5 Report

1.     Some figures are with small and almost illegible legends on the graphics

2.  “Metal-organic framework materials (MOFs) are new photocatalysts that are rapidly becoming a research hotspot. MOFs have excellent adsorption performance due to their large specific surface area and many active adsorption sites,.. “ This could be added some work, such as Inorganics, 10(2022) 202 and Micropor. Mesopor. Mat, 341(2022) 112098.

“The pollution of some toxic organic dyes to the water environment has become increasingly serious with the development of organic dyes in all walks of life. This could be added the current results, including CrystEngComm, 2022, 24, 6933–6943; Mater. Today. Commum., 2022, 31,103514; CrystEngComm, 2022, 24, 7157–7165 and J. Alloy. Compd, 2022, 897, 163178.

4. Please give the BET data for the MIL-53(Fe)@CF

5. The orbital energy of the semiconductor is very important, but in this paper, the author only calculate the Eg by the ultraviolet diffuse reflection. And it is recommended to supplement the orbital energy of the conduction band and valence band.

6. One of the most important factors affecting the degradation performance of photocatalysis is air humidity, Why did the author consider the effect of pH?

7. EPR are characterized used to explain the photocatalysis mechanism.

8. Please also provide the Raman data for checking the full characterization.

Round 2

Reviewer 2 Report

The authors have endeavored to carry out all suggestions. Therefore the manuscript may be accepted for publication.

Author Response

Thank you for your recognition.

Reviewer 3 Report

Please check which quantities should be plotted on the axes of the individual images. In the main text it is figure No. 5 (degradation degree instead of degradation rate) and all spectra in the supplementary file (wavelength instead of wavenumbers - it is other spectral method the infrared or the UV-vis spectroscopy!).

After these corrections I will be able to recommend the manuscript for publication. 

Reviewer 5 Report

accepted.

Author Response

Thank your for your recognition.